# “There Is Method to This Madness” A Qualitative Investigation of Home Medication Management by Older Adults

**DOI:** 10.3390/pharmacy11020042

**Published:** 2023-02-23

**Authors:** Olajide Fadare, Matthew Witry

**Affiliations:** Department of Pharmacy Practice and Science, University of Iowa College of Pharmacy, Iowa City, IA 52242, USA

**Keywords:** qualitative, medication, safety, self-management, cost, adherence

## Abstract

Objectives: This paper explores (1) the systems and processes older adults use to manage medications at home, and (2) the well-being goals of personal interest that motivate them. Methods: Qualitative interviews were conducted in the homes of 12 older adults in a small city in the Midwest United States. Interviews were analyzed using inductive template analysis. Results: The average age of older adults in this study was 74.2 years (SD = 10.5), 66.7% were women. The most prominent home medication management tools used were pill boxes, containers and vials, and medication lists. Routines were often aligned with activities of daily living such as teeth brushing and eating. Their medication management work occurred in contexts of other household members and budget constraints. Routines and practices were sometimes idiosyncratic adaptations and supported goals of maintaining control and decreasing vulnerability. Conclusion: In developing routines for home medication management, older adults developed systems and deliberate processes to make sense of their medication experiences in the context of their home environment and based on available resources.

## 1. Introduction

Polypharmacy (taking multiple medications) to treat chronic conditions is more common as adults age [1]. On average, older adults take 4–5 medications at a time, mostly for cardiovascular diseases, depression, and pain [2,3]. While medications can have therapeutic benefits, polypharmacy is associated with a significant amount of morbidity, mortality [4,5], and potentially avoidable health care utilization [6,7,8]. For example, a five-year longitudinal study by Chang et al. (2020) observed over 2 million (67%) polypharmacy-related hospitalizations among the sample of older adults studied [9]. Polypharmacy also increases the risk of accidental and intentional drug poisoning [1,10]. While medications are a mainstay of medical treatment for older adults with chronic conditions, when someone goes home from the hospital, clinic, and pharmacy, they are largely left to develop their own routine and system for medication management and organization. As such, decisions, practices, adaptations, and supplementations that shape patients’ home medication use and management often go unreported and undocumented [11,12,13]. Although doctors, pharmacists, and other health care professionals can have a positive impact on people taking their medications as prescribed, ultimately it is up to the patient to make decisions about what medications to take (or not), when, and if they modify or supplement their regimen. Moreover, prescriber and pharmacist guidance on communicating with patients about medicines typically includes inquiring about adherence but often does not include asking patients about their medication management and medication-taking routine, even though there is evidence that a patient’s ability to articulate their medication-taking routine has a positive association with adherence [14].

Some studies have examined the medication management systems of people in their homes using a work-system safety paradigm. The work-system safety paradigm conceptualizes the residential home as a system within which patients carry out the self-healthcare task of medication management [15,16]. For instance, Holden et al. (2017), using a human factors approach, conceptualized home medication management as patient work that is executed within a patient work system. These authors conceptualized the patient work system as comprising a microdomain of the patient interacting with tools and technology to carry out medication management tasks within the home, nested within a macrodomain of social, physical, and environmental factors [17]. According to Doucette and colleagues (2017), the systems approach to home medication management (SAHMM) model, the patient’s work system, comprises six components: (a) the patient, (b) medication management tasks, (c) medication management tools and technology, (d) the internal layout of the patient’s home, (e) availability and access to healthcare providers within the area, and (f) the social context or living arrangement within the home, that is, whether the patient lives with family and caregivers, or lives alone. These six components interact to shape the home medication management processes that influence patient outcomes [18]. Other studies have also suggested that patient outcomes such as medication adherence and medication safety are influenced by factors beyond the perimeters of conventional healthcare systems such as hospitals and pharmacies, and many of these influencing factors pertain to the strategies that patients use to manage their medications at home [15,16,18,19,20,21].

Strategies for home medication management include medication sorting and organization [22], establishing routines and cues for medication-taking such as associating it with teeth brushing and other activities of daily living [23], and requesting prescription refills [16]. Available evidence suggests that these strategies may facilitate patient adherence to treatment depending on their respective implementation contexts [24]. However, understanding why these strategies work, and the underlying cognitions that inform how patients perform medication management at home, is lacking. Further, as medication adherence and safety are goals that are typically defined and framed from the perspective of healthcare professionals, we propose that patients also have personally important well-being goals that drive patients’ engagement in home medication management. Exploring the lived experiences of community-dwelling medication users may offer new insights into their home medication management processes and cognitions, and the well-being goals of personal interest that motivate them. Such insights, together with other studies, may help primary care providers, pharmacists, and other stakeholders identify and develop potential service, product, and system targets for improving the quality and safety associated with using medications at home.

The objectives of this study were to use the systems approach to home medication management model (SAHMM) (Figure 1) to (1) describe the home medication management system and processes for a sample of community-dwelling older adults, and (2) to elucidate the well-being goals of personal interest that motivate their engagement in home medication management.

## 2. Materials and Methods

This qualitative analysis was part of an intervention study of a medication management questionnaire designed to help pharmacists make recommendations to patients about home medication use. The study was approved by the University of Iowa IRB. Participants were patients of a local independently-owned community pharmacy in a small city in the Midwest US. The pharmacy staff offered participation in the study during May and June of 2018 to patrons who were 55 years or older and took four or more medicines. The pharmacy primarily recruited during less-busy times when the study could be explained and the questionnaire could be administered, thus, convenience sampling. The rationale for these criteria was to target persons expected to have developed routines for taking their chronic medications who were either nearing retirement or retired but still managing their own medicines. Patients completing a questionnaire could mark if they were interested in being contacted for an in-home interview about how they manage their medications at home. Interviews were expected to last about 60 min and participants were paid $50 for their participation in the interview. Twenty-four patients agreed to participate in the questionnaire study and of these, 12 also agreed to be interviewed. Informed consent was obtained from each participant, including permission to share photographs of their medication organization system in published research reports. Interviewing 12 participants generally is sufficient for saturation for studies in health services research [25].

Interviews took place in participants’ homes and primarily occurred around the kitchen table, although participants also would lead the investigator around the home to show the various places where participants engaged in medication-related work such as medication storage. Most interviews were one-on-one, with a small number also including brief contributions from participants’ spouses or caregivers. Interviews were conducted by a Ph.D. investigator with training in qualitative research methods and 10 years of qualitative research experience. The investigator’s positionality included being a health services researcher and educator with expertise in patient-centered communication and medication adherence behaviors. The interview guide is included in Appendix A. The general interview topics included the medication management approach for a typical day, routines and cues, organizers, storage, difficulties, tools used, lists, and approach to refills. Other topics discussed were perspectives on physician and pharmacist roles in medication management. Participants consented to photos being taken of key medication organization setups, and names and other identifying information were blurred for privacy. Interviews were audio-recorded and edited to remove any identifying information, and transcribed by an online transcription service. Data were managed using MAXQDA v2018, Verbi Berlin.

### Data Analysis

Interviews were de-identified and analyzed after all the interviews were conducted and transcribed. Transcripts were analyzed using template analysis for descriptive coding based on the SAHMM complemented with inductive qualitative analysis to capture insights into the responses that were beyond the descriptive codes of the template [26]. In this process of descriptive coding, the researchers sought to present the facts of observation as seen through the eyes of the study participants. This was carried out by retaining the study participants’ own words through data analysis and reporting, as evidenced by the representative quotes highlighted in the results section. The SAHMM was used as a template to provide initial code categories for data vignettes. Areas of congruence of discrepancy between the theoretical propositions of the SAHMM and the study data were explored during data analysis. The two authors worked iteratively and collaboratively to apply several rounds of descriptive coding using a constant comparison approach of reviewing and coding the transcripts independently, getting together to reconcile codes and themes, reviewing the emergent themes against the relevant literature, and going back to the transcripts. Themes were identified across transcripts based on the conceptualization by Boeije (2020) of being short interpretive statements that capture salient ideas that weave across subjects and dialogue [27]. Representative quotes were identified to demonstrate the voices of a variety of participants and their experiences. To support the trustworthiness of the analysis, the authors offer a positionality statement, used a conceptual model for template analysis, engaged in frequent discussions about interpreting transcripts, coding, and themes, and used pictures and representative quotes to support the overall narrative of the analysis.

## 3. Results

Twelve older adults were interviewed in their homes. Nine were women and the sample had an average age of 74.2 years (SD = 10.5) (Table 1). The average number of medications used was 7.75 (SD = 3.5). Most medication storage and manipulation occurred in the kitchen, with some participants also storing medications in the bedroom and bathroom. Most participants also used various types of 7-day organizers. Two participants took their medication doses directly from the prescription vials.

Template analysis of the study transcripts revealed three components for the work system of older adults engaging in home medication management. These were Person, Tools and Technology, and Household. Two dimensions of home medication management process—patient work and collaborative work—were identified. Two themes—maintaining a sense of control and avoiding vulnerability—were used to describe the personally important well-being goals of older adults managing their medications at home. A theme—“there is method to this madness”—was used to describe the cognitions that underpin home medication management. The latter theme emerged from the inductive analysis of the data as informed by the literature related to cognition and patient work. Themes and representative quotes are summarized in Table 2.

### 3.1. The Home Medication Management Work System

Three components of the patient’s home medication management work system emerged from the qualitative interviews and were described using the SAHMM model. The three work-system components were: Person, Tools and Technologies, and Household.

#### 3.1.1. Person

The Person component of the home medication management work system refers to the individual who is responsible for medication management in the home. Most study participants reported living with a spouse and considered their health to be satisfactory. The older adults in this study were generally able to perform routine activities with little to no assistance and were actively involved in their home medication management, although some spouses assisted. Two characteristics were particularly relevant from the interview data under the Person domain—being a long-time medication user and admitting to occasional memory or forgetfulness issues (Table 2).

#### 3.1.2. Tools and Technology

The most prominent tools used by these older adults for home medication management were pill boxes and containers (Figure 2), medication lists, and pill cutters. The participants in the present study used a variety of pill boxes based on a 7-day compartment design, some used 7-day boxes with several dose containers per day (e.g., morning and evening) or multiples of the 7-day organizers. Some also used additional travel containers or multi-compartment vials for extra supplies. Some repurposed small dishes to set out the next dose or a special dose, and one participant’s spouse repurposed 2 days of a 7-day organizer for morning and evening supplements which were refilled daily. A common practice was to write on the containers with permanent markers, although these tended to fade or wear out over time.

In addition to pill boxes, these adults often devised their own approach to creating and managing a medication list. Information managed with medication lists varied but included descriptions of the medications and doses, and some lists contained additional information such as pharmacy information, medical conditions, and allergies.


*“…that’s [my medication list] in my purse… the first time the [pharmacy staff] printed it out for me.”*
2W [medication list]


*“I took the information off of the bottle, the pharmacy bottle [and put them in a grid] … I just did that [the grid] myself with a ruler [on paper].”*
7M [medication list]

Pill cutters were used to divide tablets in half to aid in swallowing


*“…having trouble with swallowing some of the pills so I use a pill cutter.”*
3M [pill cutter]

Older adults used technology mostly for finding out things such as medication identity and side effects. Technologies used were the internet, computers and word processors, and stationery.


*“if I Google [check the internet for] the number [on the pill], I can find out what the med is”*
5W [internet]

#### 3.1.3. Household

The household component of the home medication management work system refers to the physical, social, and economic attributes of the home that influence medication management. For the older adults in this study, these household attributes were physical locations and spaces within the home for medication storage, other household members (e.g., spouses, children, and siblings), and cost (economic) concerns.

Medication storage spaces were distributed across multiple locations within the home and commonly included the kitchen, where many older adults filled their weekly pill boxes, stored medications in cupboards, and took their medications from their organizers; the bathroom, where many stored extra medication or as-needed medications in a medicine cabinet; and the bedroom, where some used drawers to store their medications or stored a large plastic tote with their extra medications and supplies). See Figure 3.

It also was common to intermix medications with other household or personal items, whether next to glassware in the kitchen or a dresser drawer alongside personal items, sometimes the interviewee intentionally created a dedicated storage space, but that was not the norm. [storage spaces].

Another finding under household was the role of other household members and how they influenced patient work of medication management. Family members often interfered in the patient’s home medication management by encouraging, but occasionally disrupting patient routines.


*“She [my wife] reminds me to take them [my drugs].”*
11M [Others in household] [routine cue]


*“Oh my husband will sit there and he says, “Did you take your pills?”*
5W [Others in household] [routine cue]

Last under household were concerns about the cost of medications given the impact of cost on a household budget. This included medication co-pay amounts, out-of-pocket costs for provider visits, and the cost of procuring additional pharmacy services to aid home medication management. Cost concerns were sometimes substantial enough to make older adults anxious about maintaining their medication supply long term, although none admitted to an acutely critical situation, one did mention the hypothetical of choosing between paying for their medications or other essential household expenses. Moreover, purchasing supplemental pharmacy services to aid home medication management appeared to be limited by cost.


*“My husband has a medication that’s costing him a hundred and fifty bucks every time he refills it. That’s out-of-pocket. That’s not what the insurance company kicks in behind him. Really?”*
9W [cost concern]


*“…you fall into a hole [donut hole; a coverage gap in most Medicare drug plans]… Do I want to pay for my [meds]? Well, living in Iowa, you know how cold it was last winter? It was cold. Do I want to eat? Do I want to stay warm? I won’t take my meds.”*
11M [cost-concerns]


*“I think $5 a week would be pretty expensive. Or $10 a week would be expensive. That would be $520 a year. I could handle $120 but not the $520. So I could call and find out and maybe start the pill thing. That would be pretty convenient. I think I need help. I really do. It’s just getting very confusing sometimes. Especially if I run out of pills and I don’t have them, or vacations. I wish that the pill thing [pharmacy prescription medicine pre-packaging service] would be for free.”*
12M [cost-concerns]

### 3.2. Home Medication Management Processes

Home medication management processes refer to how patients managing medications at home allocate resources and effort toward the achievement of desired health outcomes. Home medication management processes emerge from patients considering all the components of the work system simultaneously, rather than as independent factors, in their judgment about how ‘best’ to manage their medications at home. Consistent with the SAHMM model, two dimensions of home medication management processes—patient work and collaborative work—were observed in this study.

#### 3.2.1. Patient Work

For this sample of older adults, patient work included several common activities such as procuring medications, organizing medications and filling pill boxes, taking medications, and monitoring their medications. Many of these activities were based on a routine they had established.

For medication procurement, participants typically called the pharmacy or signed up for automatic refills.

Participant 8W said “*when I get down to five tablets, I call [the pharmacy] to refill”, while participant 4W said*, “*[the pharmacy] keeps me well supplied [medication procurement]*”.

Participant 7M used a combination of manual and automatic refills; “*first of all, [the pharmacy] has got me on an automatic renewal… if it’s going to be a weekend or holiday, then I’ll check my medications especially the [insulin] and if it looks like I will run out, I’ll call [the pharmacy] and they’ll get me fixed up*.” 7M [procurement]

The sorting and filling of medication boxes was a prominent activity that built on their choice of tools, most commonly, some version of the 7-day pill organizer, but sometimes a less conventional approach. Some older adults also repacked their medications into other containers. See Figure 4.


*“Like I go through mine and I make sure that I have … Well, there’s a half of pill that I take for my heart. I make sure that’s in there. I make sure my [brand name medication] is in there. I make sure my metformin is. I take one blue pill. I put them in a little dish and kind of spread them apart and I take three of these. I make sure there’s three of those in there.”*
12M [medication sorting]


*“I think I portioned them all out and then I just take whatever … I don’t have the original bottles on those. I just de-cap them, I’d say. But I could tell you probably what they are, but I don’t take those very often.”*
6W [repackaging]

Once these adults had procured and organized their medications, they then move into their established routine of medication-taking. The routine for taking their medication often was aligned with activities of daily living such as teeth brushing and eating [routine cue], with some adjustments needing to be made for special circumstances such as recreation and travel [routine disruption].


*“When I get up, I take my medications where I brush my teeth.”*
4W [routine cue]

Lastly, these individuals also monitored medications, especially for side effects, and to detect potential lack of fit between dosage recommendations and activities of daily living. They may then act on this information by changing how they take the medication or by seeking medical advice. Several participants also were savvy in verifying the name and physical characteristics (e.g., shape and color) of medications.


*“sometimes you have to communicate with your physician when they give you a prescription and it might be a case of doing more harm than good. He started me out originally on one kind of high blood pressure medicine and I realized that I was coughing all the time and it was giving me a chronic cough. I finally said I don’t think this one’s a good one for me. Then he switched it and then I stopped coughing.”*
4W [monitoring] 


*“as we get older, we get confused about our meds and especially if you have two that look very similar to each other. I know to look on them and there’s different numbers on there, then I also know that if I Google the number, I can find out what the med is.”*
5W [verification]

#### 3.2.2. Collaborative Work

Patients’ medication management in the home often involves collaborations between the patient and healthcare professionals. These relationships allow health care professionals to assist patients in sensemaking about their medications and conditions and monitoring their treatment regimens. The professionals most frequently consulted by the participants in this study were physicians and pharmacists. Overall, most of the participants had positive relationships with their physicians. Several participants expressed confidence in their physicians’ orders and adherence to their prescribed regimens. Some patients expressed a preference to be on fewer medications and appreciated efforts to deprescribe.


*“Well, and I think, you know, docs [doctors] can fall into ruts, too. I mean, you know, I’ve not ever had a bad doctor, but you know, they get into routines, so they just, you know, “Oh, it’s Mary. Okay. Well, she’s … yeah. Yeah. She’s still overweight. Yeah, she’s still got… oh, yeah … Her bloodwork’s good. Okay. Renew her meds. Done,” without thinking, “Oh, she’s taken that med for how long?” You know, that kind of thing.”*
9W [medication inertia]

Like interviewees’ positive feelings for their personal physicians, study participants who were also patrons of the participating independently owned pharmacy had developed valuable personal relationships with the pharmacy staff. Trust and familiarity allowed study participants to feel comfortable collaborating with the pharmacy for medication management in an ongoing fashion. Personable communication from the pharmacy staff also fostered patient–HCP collaboration.


*“To me, a pharmacy is a trust issue. I don’t know what it’s like to go to [chain pharmacies]. I go to [my pharmacy], and I’ve gone there for 40 years. Why would I change?”*
9W [trust]

Overall, the generally positive experiences and perspectives of study participants toward their doctors and pharmacists likely served as a relevant backdrop for how older adults take their medications at home, and the systems and routines they have created for their medication management.

### 3.3. Outcomes: Personally Important Well-Being Goals of Home Medication Management

The perspectives and behaviors of the older adults in this study suggest their medication management practices appear to be in pursuit of personally important well-being goals as described by two themes—maintaining a sense of control and avoiding vulnerability. This desire for control surrounds many of their decisions and routines related to how these older adults used and managed their medications and interacted with their health care providers.


*“…they [insurance plan] were pretty insistent that they wanted to do mail-order meds. And I finally yelled and said, “Do I have to change companies? How many times do I have to say ‘no’?” To me, that doesn’t make sense. I need to deal with somebody locally because if I’ve got a question, I want to be able to call. I like knowing where my meds come from.”*
9W [control over medication supply]

Patients also engaged in medication management as a way to avoid feeling vulnerable. Avoiding vulnerability informed patients’ adherence to treatment as medication-taking was viewed as a way to control events and avoid hospital visits, emergencies, or general downturns in health.

### 3.4. Interpretation across the SAHMM Domains: “There Is Method to This Madness”

Older adults doing the work of home medication management did not necessarily follow a simple prescriptive set of activities. Rather, patients developed their own home medication management practices based on an appraisal of the resources that were available and accessible to them, and how these resources intersected with well-being goals that were important to them. Home medication management practices that may appear random, haphazard, and sometimes counterintuitive [madness], were clever adaptations of common structural elements of the work system described above in ways that allowed each patient to achieve their desired medication management outcomes [method]. Previous research suggests that patients engaging in home medication management use cognitions to understand their medication experiences, and to form connections between the attributes of their work systems, their activities of daily living, and their health. Given the available resources and their health and medication experiences, older adults used cognitions such as sensemaking to develop home medication management routines and to inform collaborative work with their primary care providers. This included making connections between past medication experiences and a current situation and considering the potential outcomes of different medication use actions. This relationship between medication experiences and outcomes informed how older adults developed their routines and adapted tools and technology to perform medication management tasks.

*“That might be my husband’s. Yeah. That’s why it’s [medication container] upside down, ‘cause it’s his. That’s his way, he knows that he took it. The next time he takes it, he flips it up. It’s his little system. I just leave it alone.*”5W [there is method to this madness]

## 4. Discussion

The study findings show that home medication management is a complex work, with deliberate, although sometimes idiosyncratic, aspects. These routines appear to be performed by patients motivated by well-being goals of maintaining control and decreasing vulnerability. The home is a medication management workplace where patients use various tools and technologies, combined with household social and economic considerations, to integrate medications into daily life.

When community-dwelling older adults return to their homes with medications from the clinic or pharmacy, they assume responsibility for managing their medications and taking them as prescribed. However, this duty is fulfilled differently by every individual and there were few consistent approaches. This is consistent with other studies considering the complex sociotechnical system that shapes the lived experience of its residents [20,28]. Here, older adults managing medications at home worked to achieve a balance between the household contexts of other family members, social support, cost considerations, and others and the tools and technical aspects of medication management [16,18,28]. This balancing act may involve actions such as reorganizing the physical layout of places within the home (e.g., the kitchen, dining area, or bathroom medicine cabinet to allocate space for medication storage, redesigning routines of daily living to accommodate medication use, associating medication use with specific activities of daily living, and retrieving and organizing medication-related information, and others [16,18,23]. Sometimes this may include potentially unsafe practices such as putting medications and vitamins into dishes or other containers. These home medication management activities are not performed arbitrarily, rather, they appear to be conducted intentionally to integrate their medications into their home life, while taking into consideration the resources, constraints, preferences, and goals of home medication management.

As shown by the method-to-madness theme, older adults were actively trying to figure out medication management systems that work. In doing this, patients appeared to use deliberate, cognitive processes such as sensemaking to understand and form connections between their medication experiences, the home, and their health to guide medication use. Older adults’ sensemaking of their medication experience at home appeared to be driven by their perceived outcomes of different medication use actions beyond clinically defined medication therapy goals. For instance, adherence to primary care provider recommendations when managing medications at home, unsupervised, may depend on whether the medication use action gives patients a sense of control or makes them feel less vulnerable concerning their health. As the psycho-emotional burden of managing chronic illnesses is associated with a sense of increased vulnerability and loss of control over one’s health and well-being, with negative consequences for patient health outcomes [29,30], it can be expected that patients will seek opportunities to engage in health behaviors that may restore a sense of control to them or allow them to minimize or avoid vulnerability. These health behaviors can include refusing to take certain medications or self-adjusting one’s dosage and dosing schedule. More so, patients tend to double down on taking their medications as prescribed, particularly when the consequences, real or perceived, of not taking their medications heighten their sense of vulnerability [31,32,33]. However, while self-adjustment of medication dosage may give the patient a sense of being in control or less vulnerable, it also raises the risk of harm when done without the professional guidance of health care providers. Nonetheless, having a sense of control and avoiding vulnerability appear to be well-being goals of personal importance that may influence how older adults manage their medication at home, and warrant more attention from primary care providers and researchers [29,31,33].

At the same time, several patients had a feeling of taking too many medications and engaged with their primary care provider to reduce the number of pills they were taking, which in some cases resulted in deprescribing, which those patients appreciated. Deprescribing is the systematic identification, reduction, or discontinuation of the number and dose of inappropriate medications by healthcare providers to manage polypharmacy, reduce the risk of medication-related adverse events, and improve clinical and well-being outcomes for the patient [34,35,36]. Some older adults acted on this perceived need for deprescribing by steering the prescriber–patient communication towards deprescribing, thus creating more opportunities for meaningful healthcare provider communication with patients.

### 4.1. Implications for Primary Care

These study findings can inform health care stakeholders including family physicians, nurse practitioners, physician assistants, and pharmacists, that patients are not just passive recipients of healthcare, but are actively working as best able to manage their medications in pursuit of well-being, control, and decreasing their vulnerability associated with their medication burden [37,38]. This is underscored by older adults in this study who generally viewed themselves as ‘experts’ on home medication management [“I ask the pharmacist, “Does this interfere with anything else I’m doing?” So, I feel like I’m smart about taking [my] medicines.”] This ‘expert patient’ perception probably explains study participants’ hands-on approach to carrying out home medication management tasks and maintaining home medication management processes, especially when changes in the physiological measures of chronic disease management such as blood glucose and blood pressure prompt dosing changes that lead to alterations in-home medication management routines. Hence, providers who endeavor to engage patients in discussions about the fit between a treatment regimen and the patients’ routine of daily living, and whether the patient anticipates any difficulties in being able to effectively manage the prescribed medications at home may see greater patient buy-in. Such conversations can humanize the provider–patient relationship and may engender better patient adherence to treatment [38,39].

Further, study findings suggest that clinically defined medication therapy outcomes such as medication adherence, quality of life improvement, and avoiding adverse events are necessary but may not be sufficient to improve and maintain patients’ adherence to treatment. Making medication therapy more collaborative and focused on the outcomes of personal interest to the patient has been shown to be beneficial [37]. Patient–provider communication sessions provide ample opportunities for doing this. Beyond asking about a patient’s history of personal illness, primary care providers could increase their focus on elucidating the patient’s intrinsic motivations for engaging in home medication management and tailor their counseling accordingly [39].

Study participants also gave suggestions for human factors improvements to the design of home medication management tools and technologies. The tools and technologies that appeared to need the most improvement were medication lists and pill boxes. Most study participants used a medication list to organize information and aid in remembering. Medication lists were majorly prepared by the patients as they generally found similar artifacts (e.g., medication information sheets) received from the clinic or pharmacy confusing [“I don’t even look at the handouts. Sometimes I get screwed up. They don’t make sense.”]. Pharmacists could more routinely discuss medication lists with patients, and when possible, be involved in designing and maintaining comprehensive and up-to-date medication lists, as well-designed medications lists could facilitate prompt identification of risk of medication-related adverse events and prompt initiation of deprescribing, as may be necessary [40,41]. Patients also suggested using different colors for day and night compartments in the design of pill boxes. The concerns about medication lists and pillboxes suggest the need for more human factors considerations in designing medication management tools and technologies [42].

### 4.2. Limitations

This was a convenience sample of 12 older adults in a single Midwestern US town. While there appeared to be variation in income and education, all the study participants were white and used a single independently-owned pharmacy. Interviewing more adults, including those from other socioeconomic and demographic groups, may yield additional medication management practices, beliefs, and experiences. The present analysis may not have reached saturation of the diverse concept of medication management practices in the home but may serve as a start to further develop an understanding of this phenomenon. Additionally, the authors did not engage in member-checking.

### 4.3. Future Research

In addition to purposefully interviewing more demographically and socioeconomically diverse participants, future work can test how best to collect information about home medication management practices and vulnerabilities, best practices for discussing medication management practices in an open and non-judgmental way, focused on finding solutions, and what role pharmacists home visits may have in supporting older adults with their home medication management.

## 5. Conclusions

This sample of older adults had well-developed routines for medication management and use, although elements of their practices may put them at risk for medication misadventures. In developing routines for home medication management, older adults used cognitive processes to make sense of their medication experiences and available resources to guide medication use actions to avoid vulnerability and gain a sense of control in determining their well-being.

## Figures and Tables

**Figure 1 pharmacy-11-00042-f001:**
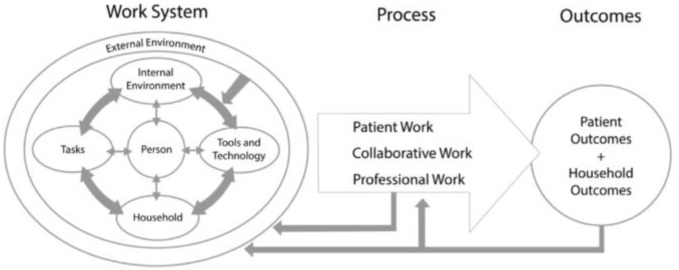
The systems approach to home medication management (SAHMM) model Reprinted with permission from Ref. [18]. 2017 Willam Doucette.

**Figure 2 pharmacy-11-00042-f002:**
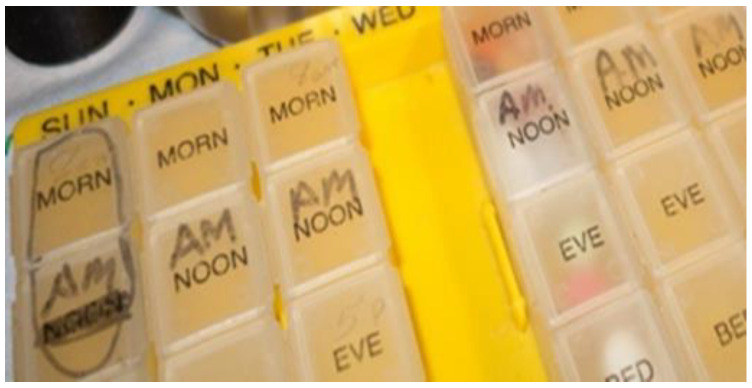
Home medication management tool—pill box with markings.

**Figure 3 pharmacy-11-00042-f003:**
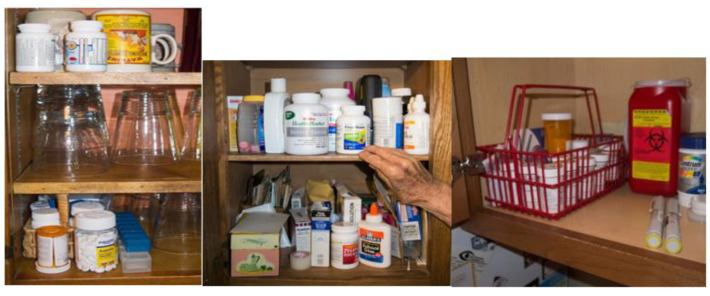
Medication storage spaces within the home.

**Figure 4 pharmacy-11-00042-f004:**
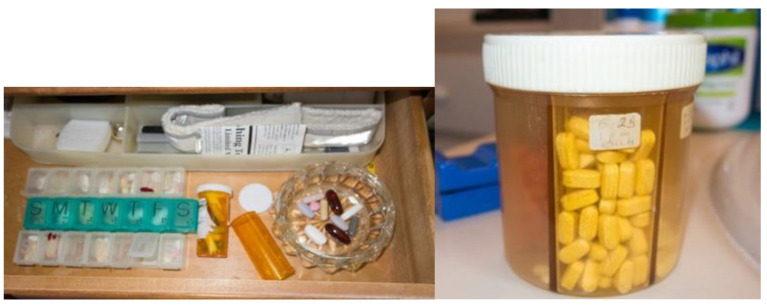
Medication storage space with loose tablets and capsules (**left**) and repurposed multicompartment vial (**right**).

**Table 1 pharmacy-11-00042-t001:** Participant characteristics.

ID	Age	Gender	Oral Medications	Storage Location and Process	Pillbox/Med Organizer
1W	91	W	5	Kitchen: General medication shelf in the kitchen area; uses dining table to sort meds and fill pillbox Bathroom: Medicine cabinet	Yes; 7-day organizer with AM/PM compartments
2W	68	W	7	Kitchen: dedicated tub Bedroom: on nightstand	Yes; 7-day organizer with AM/PM compartments
3M	84	M	7	Kitchen: Counter-top and dining table Bathroom: Medicine cabinet	Yes; 7-day organizer and a second filled two days at a time for AM/PM supplements
4W	63	W	4	Bathroom: Medicine Cabinet—stacks all medications in the cabinet with the current medications placed in front for easy reach	No; takes medications directly out of the vials
5W	62	W	5	Kitchen: Dedicated medicine basket kept on a counter-top	Yes, Both bottles and pillboxes, and sets out doses
6W	89	W	10	Kitchen: top of microwave, uses repurposed bottles and dish with pills Bedroom: Drawer	Yes; 7-day organizer with AM/PM compartments; uses 4
7M	74	M	10	Kitchen: Dedicated cabinet	Yes; 7-day organizer with AM/PM
8W	85	W	13	Kitchen: Counter-top Bedroom: Drawer, in a clear plastic bag	Yes; 7-day organizers with AM/PM compartments; uses 2
9W	70	W	6	Kitchen: Dedicated Cabinet, uses table weekly to sort and fill organizer; Bathroom: stores additional medications	Yes; 7-day organizer
10W	74	W	6	Kitchen: Counter-top Bathroom: Dedicated cabinet, with separate sides for morning and evening medications in the cabinet	No; takes medication directly out of vial. Uses pill box only when traveling
11M	64	M	5	Kitchen: Cupboard—keeps medications in a vial; refills pill boxes from vials weekly	Yes; 7-day pill organizer; uses 2
12M	66	M	15	Kitchen: Dedicated plastic bag placed on counter-top	Yes; 7-day organizer with 4-daily doses

**Table 2 pharmacy-11-00042-t002:** Concepts, codes, themes, and representative quotes describing home medication management of older adults.

Home Medication Management Components	Quotes
Person	Older adults aged 55 or older who are long-term medication users for managing one or more chronic conditions. “*remembering to take them [medications] is probably the hardest thing I have just ‘cause I get sidetracked and I’ve noticed I don’t remember things as well, so I guess that’s why I would say I’m getting older and easily distracted.*” 5W [forgetting]
Tools and Technology	“*I try to update my medication list if I’m doing something different or if they’ve added something to my [medications].*” 6W [Medication list] “*I had one off of the computer, but my computer word processor’s not working right now…*” 7M [medication list-updating] “*I like to Google things on the internet and I looked to see what any side effects would be.*” 10W [internet]
Household	“*the medicine cabinet is in the bathroom. So, let’s go to the bathroom first. I also have some in the kitchen. Those are the ones I normally take once a day or twice a day.*” 10W [storage location] “*It’s like when our kids are here, all of a sudden, our whole routine was helter-skelter, but we got it all worked in.*” 3M [other household members] “*I would hate to be … and not be able to use meds every day because of the cost. There’s several times I thought, “Well geez, I’m gonna go every other day.” And then I thought, and she [my wife] says, “No, you’re not.*” 12M [cost concerns]
Patient work	“*As we get older, we get confused about our meds and especially if you have two that look very similar to each other. I know to look on them and there’s different numbers on there, then I also know that if I Google the number, I can find out what the med is.*” 5W [verification]
Collaborative work	“*I’ll tell you what. This [number of medications] is ridiculous… He’s really tried to get me off a couple of them for me.*” 12M [taking too many medicines] “*I feel like I get support from my pharmacy on managing my meds and from my docs too. I think we’ve got a good exchange going. Establishing a relationship with the patients. They’re not afraid to ask questions or feel like taking their time. I think that just human interaction really lays a basis for the information flow.*” 3M [personable communication]
Theme: “There is method to this madness”	“*I always put my morning ones over here. I fill them in the evening … This’ll be for … I took my morning ones, this is for after supper. Then, I’ll refill before my nighttime. This’ll be for my nighttime. They’re all in different containers, so I know I’m getting the right thing. I take an anti-biotic, first thing in the morning. The way I remind myself is I leave the glass on the counter that we use for our last meds of the day, so the glass is still there when I get up in the morning. Its still sitting there and it reminds me to take my anti-biotic. There’s method to this madness.*” 3M “*Here is a new one. Gabapentin. This says, “Take one tablet my mouth three times a day.” So, okay. That to me means morning, noon, and night. But I don’t take any afternoon pills because I forget them all the time. So I take two in the morning and one at night.*” 7M “*I get up [in the morning], I’ll have my coffee and then I’ll decide to eat something and then at that time I try to take the pills at the same time. Where I get off is like I said, if he [my husband] pours my cup of coffee and I don’t go to the cupboard to get the cup then I don’t get triggered sometimes.*” 5W
Theme: Maintaining control	“*She [my physician] told me when she gave them [the sleeping pills] to me that I could take two, and I don’t allow myself to do that. I can get up on five hours of sleep and/or six and do a day. It doesn’t matter, so I’m not going to let these guys have too much control of my life, but they do help.*” 6W
Theme: Avoiding vulnerability	“*it [taking too many meds] makes me feel like I’m a sick person. Medically, I’m pretty sick. I used to be on, like, 15 prescriptions so Dr. [mentioned doctor’s name] got me off a couple of them and he’s tried the [brand medication] and that did not work. I had to get right back on it again. I know I can’t be without it.*” 12M

## Data Availability

Data sharing is not applicable to this article.

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
