# Peer review of "“There Is Method to This Madness” A Qualitative Investigation of Home Medication Management by Older Adults"

_pharmacy, 2023, doi:10.3390/pharmacy11020042_

Round 1
Reviewer 1 Report
Title: “There is method to this madness.” A qualitative investigation of home medication management by community-dwelling older adults
Overall summary: The paper describes a qualitative study that sought to explore the systems and processes that older adults use to manage their medications at home, and identify their personal well-being goals. The manuscript is generally well-written and adds some insights to current literature regarding home medication management and how pharmacists can contribute to optimizing the medication experience for this population
The title is rather long. You may consider shortening it. May not be necessary to qualify with “A qualitative investigation”
Abstract:
- The abstract does a really good job of summarizing the paper and accurately reflects the content.
Introduction:
- The study objectives are clearly stated. Very helpful.
- Relevant literature is synthesized and summarized
Materials & Methods
- What was the rationale for the inclusion criteria of 55 years or older?
- Why were participants recruited from only one pharmacy?
- Why were 12 interviews deemed sufficient? On what basis? Please support with appropriate literature.
- Page 3; ln 113: “personality” means something different from how you use it here. Please change to a term that more aptly captures what you are referring to.
- Recommend changing sub heading 2.1 from “Interview topics” to “Data analysis”, and describing what that process was. How were participants contacted? Who scheduled the interviews?
- Recommend changing 2.2 to “Data Analysis”
- Please define and describe the analytical approaches mentioned – “template analysis”, “descriptive coding”, “inductive qualitative analysis”. The “iterative and collaborative” approach is not clear. What was this process? For example, did you code each transcript independently and then reconcile codes?
- Please include a section on “trustworthiness” and describe what you did to ensure this (e.g. member checking)
Results
- I am a bit confused - so not all domains of the work system were reflected in this data? Did you use the framework to analyze that data? This goes back to the data analysis – perhaps you can describe the data analysis in clearer detail in that section
- Some of the exemplar quotes you use in the text are the same as on table 2. That is redundant. Will recommend that you either refer to the table or use other quotes. For example - “Remembering to take them is probably the hardest thing I have just 'cause I get sidetracked and 293 I've noticed I don't remember things as well, so I guess that's why I would say I'm getting older and easily distracted” – shows up 3X in the text
Discussion
- Page 12; ln 402: Remove extra word “aspect”
- Page 12; ln 426: Remove word “used”
- Page 14; ln 504 (under limitations): What do you mean by “demographic group”?
Generally, well-written. Very wordy. Recommend editing down a bit and laying out the results in a more reader friendly manner. I had to go back and forth to know where which theme started.
Author Response
Reviewer 1
Title: “There is method to this madness.” A qualitative investigation of home medication management by community-dwelling older adults
Overall summary: The paper describes a qualitative study that sought to explore the systems and processes that older adults use to manage their medications at home, and identify their personal well-being goals. The manuscript is generally well-written and adds some insights to current literature regarding home medication management and how pharmacists can contribute to optimizing the medication experience for this population
Thank you.
The title is rather long. You may consider shortening it. May not be necessary to qualify with “A qualitative investigation”
We have revised the title for more brevity, but prefer to retain “qualitative investigation” per most guidelines that have you note qualitative in titles for qualitative studies. Our new title is:
“There is method to this madness.” A qualitative investigation of home medication management by older adults. See line 3.
Abstract:
- The abstract does a really good job of summarizing the paper and accurately reflects the content.
Thank you for your kind observation.
Introduction:
- The study objectives are clearly stated. Very helpful.
- Relevant literature is synthesized and summarized
Thank you.
Materials & Methods
- What was the rationale for the inclusion criteria of 55 years or older?
Please see Page 3; line 96 – 99 for the rationale. “The rationale for these criteria was to target persons expected to have developed routines for taking their chronic medications who were either nearing retirement or retired but still managing their own medicines.”
- Why were participants recruited from only one pharmacy?
There was only 1 pharmacy participating in the multi-component study. We mention this in the limitations.
- Why were 12 interviews deemed sufficient? On what basis? Please support with appropriate literature.
12 is a common number of participants in pharmacy. We were restricted in that of those that participated in the larger study of the home-medication management workup that only 12 agreed to be interviewed. We interviewed everyone we had access to. We do not claim complete saturation, rather, we are using the template to organize these patient perspectives. We found or data sufficient to present our findings from the interview series.
Williams, S. D., Phipps, D. L., & Ashcroft, D. M. (2013). Understanding the attitudes of hospital pharmacists to reporting medication incidents: a qualitative study. Research in Social and Administrative Pharmacy, 9(1), 80-89.
Witry, M. J., & Doucette, W. R. (2014). Community pharmacists, medication monitoring, and the routine nature of refills: a qualitative study. Journal of the American Pharmacists Association, 54(6), 594-603.
Weir, K. R., Naganathan, V., Bonner, C., McCaffery, K., Rigby, D., McLachlan, A. J., & Jansen, J. (2020). Pharmacists’ and older adults’ perspectives on the benefits and barriers of Home Medicines Reviews–a qualitative study. Journal of Health Services Research & Policy, 25(2), 77-85.
-
Page 3; ln 113: “personality” means something different from how you use it here. Please change to a term that more aptly captures what you are referring to.
Thank you for your comment. We believe that the term you are referring to on Page 3; line 113 is “Positionality”. Positionality statements are recommended by qualitative reporting guidelines. Briefly, positionality, is a self-disclosure of the researcher’s worldview and experiences and the position they adopt about a research task and its social context, and its influence on how the research is conducted, data and results are interpreted, and the outcomes postulated. Some aspects of positionality are culturally ascribed or generally regarded as being fixed (e.g., gender, race, skin-color, nationality), while other aspects are more fluid and subjective (e.g., education, professional experience, political views). The positionality statement (Page 3: lines 113 – 115) captures the aspects of positionality relevant to the present study. Positionality is one thing we do in the present study to support trustworthiness.
Some citations on researcher positionality:
Holmes, A. G. D. (2020). Researcher Positionality--A Consideration of Its Influence and Place in Qualitative Research--A New Researcher Guide. Shanlax International Journal of Education, 8(4), 1-10.
Bourke, B. (2014). Positionality: Reflecting on the research process. The qualitative report, 19(33), 1-9.
Kirstetter, Katie. “Insider, Outsider or Somewhere in between: The Impact of Researchers’ Identities on the Community based Research Process.” Journal of Rural Social Sciences, vol. 27, no. 2, 2012, pp. 99-117.
- Recommend changing sub heading 2.1 from “Interview topics” to “Data analysis”, and describing what that process was. How were participants contacted? Who scheduled the interviews?
We have revised this section to merge better with the methods section. See Page 3; lines 115-119. The interview process, including how many participants were contacted (24), is described in Section 2. Materials and Methods. See Page 3; lines 88 – 123. A detailed breakdown of researcher roles is provided under Author Contributions- Page 14; lines 527 - 529
- Recommend changing 2.2 to “Data Analysis”
Section heading has been revised as suggested.
- Please define and describe the analytical approaches mentioned – “template analysis”, “descriptive coding”, “inductive qualitative analysis”. The “iterative and collaborative” approach is not clear. What was this process? For example, did you code each transcript independently and then reconcile codes?
“Template analysis and “descriptive coding” are described on Page 3; lines 134 – 140. Template analysis, briefly, is a qualitative analysis approach that justifies starting with a-priori descriptive codes from an established theory or framework, but then also allows for interpretive coding and identification of themes that may go beyond the descriptive nature of the a-priori framework/theory.
The “iterative and collaborative” approach is described on Page; lines 135 - 137
- Please include a section on “trustworthiness” and describe what you did to ensure this (e.g., member checking)
We have edited the end of the analysis paragraph to make this easier for the reader to identify how we approached trustworthiness. “To support the trustworthiness of the analysis, the authors offer a positionality statement, used a conceptual model for template analysis, engaged in frequent discussions about interpreting transcripts, coding, and themes, and used pictures and representative quotes support the overall narrative of the analysis.”
Results
- I am a bit confused - so not all domains of the work system were reflected in this data? Did you use the framework to analyze that data? This goes back to the data analysis – perhaps you can describe the data analysis in clearer detail in that section.
The SAHMM framework was used as a template to provide initial code categories for data vignettes during data analysis. However, not all domains of the SAHMM emerged as salient for the purpose of the present study. The reporting has been updated to reflect this. See Page 3; lines 134 - 140
- Some of the exemplar quotes you use in the text are the same as on table 2. That is redundant. Will recommend that you either refer to the table or use other quotes. For example - “Remembering to take them is probably the hardest thing I have just 'cause I get sidetracked and 293 I've noticed I don't remember things as well, so I guess that's why I would say I'm getting older and easily distracted” – shows up 3X in the text
Thank you for pointing this out. We have removed redundant quotes. See lines 188 – 190; 307 - 308
Discussion
- Page 12; ln 413: Remove extra word “aspect”
The relevant text has been edited.
- Page 12; ln 426: Remove word “used”
The relevant text has been edited.
- Page 14; ln 504 (under limitations): What do you mean by “demographic group”?
race/ethnicity, disability status, others
Generally, well-written. Very wordy. Recommend editing down a bit and laying out the results in a more reader friendly manner. I had to go back and forth to know where which theme started.
Thank you, we have done a bit of editing down, including removing redundant quotes.
Reviewer 2 Report
Well done. Excellent work.
Some minor sugestions
--regarding primary care can you also comment on the role of the family doctors?
--regarding Future research can you please develop future methods?
Author Response
Reviewer 2
Well done. Excellent work.
Thank you for your kind observation
Some minor suggestions
Regarding primary care can you also comment on the role of the family doctors?
Thank you, we have added family doctor, NP, PA, and pharmacist to 4.1 implications for primary care to make this more explicit who we are talking about.
Regarding Future research can you please develop future methods?
We are actively working on applying for funding using this study to inform new practice models, tools, and resources.